# Global Trends in Phytohormone Research: Google Trends Analysis Revealed African Countries Have Higher Demand for Phytohormone Information

**DOI:** 10.3390/plants9091248

**Published:** 2020-09-22

**Authors:** Tapan Kumar Mohanta, Yugal Kishore Mohanta, Dhananjay Yadav, Abeer Hashem, Elsayed Fathi Abd_Allah, Ahmed Al-Harrasi

**Affiliations:** 1Natural and Medical Sciences Research Center, University of Nizwa, Nizwa 616, Oman; 2Department of Botany, North Orissa University, Sri Ramchandra Vihar, Takatpur, Baripada, Odisha 757003, India; ykmohanta@gmail.com; 3Department of Medical Biotechnology, Yeungnam University Gyeongsan, Gyeongsangbuk-do 38541, Korea; dhanyadav16481@gmail.com; 4Botany and Microbiology Department, College of Science, King Saud University, Riyadh 11451, Saudi Arabia; habeer@ksu.edu.sa; 5Plant Production Department, College of Food and Agricultural Sciences, King Saud University, Riyadh 11451, Saudi Arabia; eabdallah@ksu.edu.sa; 6Mycology and Plant Disease Survey Department, Plant Pathology Research Institute, ARC, Giza 12511, Egypt

**Keywords:** pytohormones, abscisic acid, auxin, brassinosteroids, cytokinin, ethylene, gibberellins, jasmonic acid, salicylic acid, strigolactones, Google trends

## Abstract

The lines of research conducted within a country often reflect its focus on current and future economic needs. Analyzing “search” trends on the internet can provide important insight into predicting the direction of a country in regards to agriculture, health, economy, and other areas. ‘Google Trends’ collects data on search terms from different countries, and this information can be used to better understand sentiments in different countries and regions. Agricultural output is responsible for feeding the world and there is a continuous quest to find ways to make agriculture more productive, safe, and reliable. The application of phytohormones has been used in agriculture world-wide for many years to improve crop production and continues to be an active area of research for the application in plants. Therefore, in the current study, we searched ‘Google Trends’ using the phytohormone search terms, abscisic acid, auxins, brassinosteroids, cytokinin, ethylene, gibberellins, jasmonic acid, salicylic acid, and strigolactones. The results indicated that the African country Zambia had the greatest number of queries on auxin research, and Kenya had the most queries in cytokinin and gibberellin research world-wide. For other phytohormones, India had the greatest number of queries for abscisic acid and South Korea had the greatest number of ethylene and jasmonic acid search world-wide. Queries on salicylic acid have been continuously increasing while the least number of queries were related to strigolactones. Only India and United States of America had significant numbers of queries on all nine phytohormones while queries on one or more phytohormones were absent in other countries. India is one of the top five crop-producing countries in the world for apples, millet, orange, potato, pulses, rice, sugarcane, tea, and wheat. Similarly, the United States of America is one of the top five crop-producing countries of the world for apples, grapes, maze, orange, potato, sorghum, sugarcane, and wheat. These might be the most possible factors for the search queries found for all the nine phytohormones in India and the United States of America.

## 1. Introduction

Phytohormones are a group of chemical substances found in the plants that regulate growth, development, organogenesis, and stress responses in plants [1,2,3,4,5,6]. Phytohormones play an active and critical role in all stages of plant development, from seed germination [7], to root [8], shoot [9], leaf [10], stem development [11], flowering [12], fruiting [13] and ripening [14,15,16]. Plant hormones function as an important signaling molecules [2] and are synthesized in a genetically programmed manner throughout the lifecycle of a plant. Each and every individual cell of a plant is capable of producing and transporting hormones to the specific tissues and organs [17,18,19,20]. Phytohormones are also present in algae [21,22], fungi [23,24,25], and bacteria [26,27,28], which often form a symbiotic relationship with the host plants or combined in the case of lichens to create a unique form of plant life. The variety of demonstrated phytohormones commonly found in plants are abscisic acid, auxins, brassinosteroids, cytokinin, ethylene, gibberellins, jasmonic acid, salicylic acid, and strigolactones. Abscisic acid is generally recognized as a growth inhibitor and was previously known as dormin/abscicin II [29,30].

Phytohormones play a crucial role in every aspect of plant growth and development, as well as stress responses. The controlled synthesis and coordinated homeostasis of these hormones regulate cellular metabolism in plants and their entire life cycle. The utilization of these hormones in agriculture and horticulture are of enormous importance and growers have been using these hormones for regulating various aspects of plant productivity since their first discovery and commercial availability [31,32]. A healthy plant is critical for maximizing yield and obtaining high quality harvested products [33,34]. Millions of hectares in the world are used to grow crops each year, and plant growth regulators are frequently used to increase the crop yields [35]. Their use increases the economic value by enhancing the quality and quantity of the crops [36]. Growth regulators can also be used to manipulate harvest dates to avoid losses in crop yield due to adverse environmental conditions [15,37]. Growth regulators are extensively used in the propagation of plants from stem cuttings, reducing the time required to generate the plants that can be used in field plantings [38]. Phytohormone treatments have an enormous potential to provide higher crop yields to address the food needs of the ever-increasing population in the world [39,40]. Good agricultural land space is limited, and the world population continues to increase, placing ever greater pressure on the ability to increase crop yields and grow crops on non-optimal land. The ability to develop a sufficient, reliable, and secure amount of food is a daunting task, and phytohormones have been used to help achieve this goal. A greater level of research is still needed; however, to achieve the most effective use of phytohormones.

Environmental, edaphic, and climatic conditions are not the same in all the locations where crops are produced. The climatic conditions vary considerably in the tropic, sub-tropic, temperate, and polar regions of the world, affecting the yields and the types of crops that can be grown. In addition, physical, chemical, biological, and anthropogenic activities (edaphic factor) of soils also play an important role in plant growth, development, stress responses, and ultimately crop yield. Currently, there are 253 countries in the world that are distributed in different climatic and edaphic regions, and among these countries, relatively few are agriculturally productive. The top five crop-producing countries of the world for various crops are mentioned in Table 1 [41].

From the above, it clearly shows that different countries represent varied climatic and edaphic regions in the world. Therefore, the cultivation of a specific crops, as well as their quality and quantities are largely dependent on the presence of a suitable environmental conditions. Therefore, some countries are highly productive, while others cannot compete with it. In addition, climatic and edaphic factors, advanced knowledge of science and the implementation of advanced technology in agriculture and research also play an important role. During the era of information technology, people (researchers, farmers, students, and consumers) want to obtain knowledge through the internet to implement this knowledge directly in agriculture and future research. Regarding the internet, the majority of computer users throughout the world use Google as the default search engine to search their queries [42,43]. Therefore, in the present study, we attempted to use ‘Google Trends’ to analyze the trends of people in different countries of the world in regards to phytohormones. Google dominates the search engines used on the internet and the term “Google” has become a routine and household word associated with internet browsing. Approximately 3.3 billion searches occur in Google each day and a total of 2 trillion Google searches were estimated for 2018 [44,45]. People use Google to search many topics, including health, practical information, online shopping, event booking, and other available online resources [46,47,48]. Therefore, Google search data provides a major advantage over traditional survey data. In addition, the use of ‘Google Trends’ enabled us to survey the weekly, monthly, and yearly use of phytohormone search queries without any cost.

This analysis will enable us to better understand which countries are querying information on phytohormones and benefitting from their search queries regarding phytohormone research and implantation in agriculture production. In addition, it also enables us to understand which countries are lagging behind and need additional support on specific topics and applications. The “Google Trends” search engine is an excellent tool for assessing country by country trends in the use of search terms searched using the Google browser. An analysis of phytohormone query trends using ‘Google Trends’ will provide considerable information about the importance of phytohormone research globally.

## 2. Results

### 2.1. Country-Wide Search Trends

#### 2.1.1. India Leads in Abscisic Acid, and Zambia Leads in Auxin Search Queries

In the present study, we searched and analyzed all the naturally-occurring phytohormones, including abscisic acid, auxin, brassinosteroids, cytokinin, ethylene, gibberellins, jasmonic acid, salicylic acid, and strigolactones. All 253 countries presently designated in the world were included in our analysis of query data from 1 January 2004 to 20 October 2019. Country-wide analysis using ‘Google Trends’ of different phytohormone-related queries by different countries revealed a spatiotemporal result (Figure 1). The countries with the greatest number of queries for abscisic acid using Google trends were India, Philippines, Australia, Canada, Thailand, the United States of America (USA), and the United Kingdom (UK) (Appendix A). A total of 53 countries were associated with queries on auxin. A few of the countries with high query values for auxin were Zambia, Ethiopia, Nigeria, Nepal, Jamaica, Tanzania, Pakistan, New Zealand, India, South Korea, Malaysia, China, Vietnam, Kenya, and others (Appendix A).

#### 2.1.2. China Leads in Brassinosteroids and Kenya Leads in Cytokinin Search Queries

Brassinosteroids are steroid hormones with C_27_, C_28_, and C_29_ carbon-containing compounds with substituted side chain alkyl groups. China was found to have the greatest number of search queries for Brassinosteroids followed by India and the USA (Figure 1, Appendix A). No other countries were found to be associated with BRs search queries as determined with ‘Google Trends’. The country with the greatest number of cytokinin search queries was Kenya, followed by Mauritius, Nigeria, Pakistan, New Zealand, India, Philippines, Australia, South Africa, Malaysia, Canada, USA, Egypt, China, UK, and Poland. No additional other countries were associated with cytokinin search queries (Figure 1, Appendix A).

#### 2.1.3. South Korea Leads in Ethylene and Kenya Leads in Gibberellin Search Queries

Among 253 countries, South Korea was found to have the greatest number of ethylene search queries, followed by Singapore, Malaysia, Qatar, India, Pakistan, USA, Canada, Nigeria, Lebanon, Taiwan, Philippines, Australia, United Arab Emirates (UAE), and others. In total, 60 countries were found to be associated with ethylene search queries (Figure 1, Appendix A). In contrast, only 15 countries were found to be associated with gibberellin search queries. Kenya had the greatest number of queries followed by Nigeria, Pakistan, Jamaica, India, South Africa, Philippines, Australia, UK, Malaysia, USA, China, Canada, and Spain (Figure 1, Appendix A).

#### 2.1.4. South Korea Leads in Jasmonic Acid and Philippines Leads in Salicylic Acid Search Queries

South Korea had the greatest number of jasmonic acid search queries followed by Iran, Japan, China, India, Germany, Mexico, Malaysia, Colombia, Canada, and others (Figure 1, Appendix A). In total, 20 countries were found to be associated with jasmonic acid search queries in Google. In contrast, 53 countries were found to be associated with salicylic acid search queries. The Philippines had the greatest number of salicylic acid search queries followed by Trinidad and Tobago, Singapore, Nepal, Canada, USA, Australia, Ireland, Ghana, UK, South Africa, New Zealand, Malaysia, UAE, India, Sri Lanka, Kenya, Pakistan, and others (Figure 1, Appendix A).

#### 2.1.5. China Lead in Strigolactones Search Queries

China had the greatest number of strigolactones search queries among the 253 countries, followed by Taiwan, Australia, Japan, Germany, India, and the USA. No additional countries were found to be associated with strigolactones search queries (Figure 1, Appendix A).

### 2.2. Year-Wise and Total Number of Search Queries

#### 2.2.1. Salicylic Acid Queries Are Increasing

All of the data on phytohormone search trends from 1 January 2004 to 20 October 2019 were downloaded to analyzed yearly trends in phytohormone search queries. The results indicated that of the number of salicylic acid search queries was higher than for other phytohormones and that the number of salicylic acid queries was increasing yearly (Figure 2 and Figure 3, Table 2, Appendix A). The mean statistical values of the Google trend result over time for abscisic acid, auxin, brassinosteroids, cytokinin, ethylene, gibberellins, jasmonic acid, salicylic acid, and strigolactones were found to be 27.56, 31.61, 15.80, 27.53, 47.68, 23.16, 18.96, 50.62, and 16.51, respectively. Salicylic acid has the highest mean of 50.62 for search query, followed by 47.68 ethylene (Table 2). However, the standard deviation of ethylene (17.00) was highest among the others (Table 2). Frequency distribution showed brassinosteroids remained more concentrated towards the range of 0–15 while ethylene, cytokinin, and salicylic acid were more concentrated towards the range from 60–100 (Figure 3). However, the frequency distribution of the individual phytohormones was quite diverse, and the occurrence of distinct values in the variable was not so abundant. This explains the dynamic search trend of the phytohormones in Google.

The greatest number of queries for the salicylic acid trend was followed by queries on ethylene. Notably, the number of ethylene queries was the highest during the beginning of 2004 and has been gradually decreasing with time (Figure 2). The number of auxin queries followed the number of salicylic acid and ethylene queries. Notably, however, the number of auxin search queries was similar to the number for cytokinin at the present time (Figure 2). The number of queries on jasmonic acid was quite low. The lowest number of search queries; however, was for strigolactones (Figure 2). While the initial number of strigolactone queries was quite low in 2004, the number has been growing and strigolactones are receiving considerable attention at present. Regardless of the increase, however, the number of queries on strigolactones is still the lowest among all phytohormone search queries globally (Figure 2). The number of auxin and abscisic acid queries has remained quite stable with intermittent ups and downs. In contrast, the number of queries on gibberellins and jasmonic acid has been gradually decreasing over time (Appendix A).

#### 2.2.2. Only India and The United States of America Had Search Queries for All of the Examined Phytohormones

A Venn diagram was constructed to compare the number of queries for the different phytohormones in different countries. A total of 15 different combinations of phytohormone search queries was examined (Appendix A, Table 3). Only India and the United States of America were found to have search queries on all nine phytohormones (Figure 4, Appendix A). As illustrated, the greatest share of queries on all nine phytohormones were associated with India and the USA (Figure 4). Search queries for one or more of the phytohormones were absent in the other countries. In addition to India and the USA, some of the countries with the highest and most comprehensive number of queries on the nine phytohormones included were Australia, Canada, China, Kenya, Malaysia, New Zealand, Nigeria, Pakistan, Philippines, South Africa, and the United Kingdom. Brassinosteroid queries were not associated with Australia, Canada, Kenya, Malaysia, New Zealand, Nigeria, Pakistan, Philippines, and the United Kingdom (Figure 4). A word cloud generated using all of the countries associated with phytohormone search queries showed India, USA, and the Philippines. Abscisic acid queries were absent in China, Malaysia, New Zealand, and Pakistan (Figure 5).

Gibberellin queries were not found for New Zealand, while jasmonic acid queries were absent in Kenya, New Zealand, Nigeria, Pakistan, and the Philippines (Figure 4). Salicylic acid queries were absent for China, while strigolactone queries were absent in Kenya, Malaysia, New Zealand, Nigeria, Pakistan, Philippines, and the United Kingdom. Queries on six common phytohormones (abscisic acid, auxin, cytokinin, ethylene, gibberellin, and salicylic acid) combined were only associated with six countries—India, the Philippines, Australia, Canada, United States of America, and the United Kingdom (Figure 4, Table 3). Queries on five common phytohormones (auxin, cytokinin, ethylene, gibberellin, and salicylic acid) were associated with Nigeria, Pakistan, Malaysia, Kenya, and South Africa (Figure 4, Table 3). The most common countries with ethylene search queries were Qatar, Kuwait, Algeria, Portugal, Colombia, Chile, Argentina, and Ukraine (Table 3). Auxin, ethylene, and salicylic acid queries were the most dominant and associated with 33 countries (Table 3).

Principal component analysis (PCA) was conducted of the countries associated with search queries on the nine phytohormones. Results indicated that India, Australia, the USA, Singapore, China, South Korea, Canada, Malaysia, Taiwan, Iran, and Japan fell distantly in coordinate 1; whereas, the United Kingdom, New Zealand, South Africa, Mauritius, Pakistan, Philippines, Nigeria, and Kenya fell distantly in coordinate 2 (Figure 6). In addition, African countries, including Kenya, Nigeria, South Africa, Zambia, and Ghana also fell in coordinate 2; whereas, no country was found to fall in coordinate 1 (Figure 6).

Clustering analysis revealed two major groups where India and China were in one group while the rest of the countries were in the other group (Figure 7). Within the latter cluster, Germany, Japan, and Iran grouped together; Austria, Hungary, Czechia, Belgium, Switzerland, Israel, Denmark, Israel, Denmark, Norway, Sweden; Kenya, Philippines, Nigeria, and Pakistan grouped together; Canada, UK, the USA, Malaysia, South Africa, New Zealand, and Singapore grouped together; and Bangladesh, Hong Kong, Sri Lanka, United Arab Emirates, and Lebanon grouped together; while South Korea grouped independently (Figure 7).

## 3. Discussion

‘Google Trends’ data have been used to monitor real-time disease outbreaks and also further used to monitor the progression of the spread as well [46,49,50,51]. Google has been used by many people to diagnose different ailments prior to going to a doctor, and there has been a surge in the number of people searching for information related to health [52]. The use of ‘Google Trends’ is equally applicable to plant science, agriculture, and other allied subjects. Therefore, for the first time, we aimed to understand the global trends associated with “phytohormone” queries. It is commonly recognized that the USA, UK, Australia, Canada, China, Germany, France, Japan, Korea, Russia, and others are high-income countries and that agriculture production is also very high. The high level of agriculture production fosters a good economy and contributes significantly towards their national GDP (gross domestic product) [53,54]. Our ‘Google Trends’ analysis of phytohormone search queries from 2004–2019 revealed some interesting results. For instance, the African country Zambia, a least-developed country with a population of 17,861,030, had the greatest number of auxin search queries among 253 countries in the world. Other African countries, including Ethiopia, Nigeria, Jamaica, and Tanzania, also led the number of auxin search queries. Agriculture in Zambia contributes 19% to its GDP and employs three-quarters of the population [55]. The major agricultural products of Zambia are cassava, maize, millet, sorghum, soybean, groundnuts, rice, and cotton. Greater knowledge about trends in phytohormone search queries for Zambia may reflect and indicate research, production, manufacturing, and employment sector needs. The African country Kenya had the greatest number of cytokinin and gibberellin search queries globally, followed by Nigeria. Agriculture represents 26% of Kenya’s GDP and provides more than 40% of the nation’s employment [56]. The major agricultural products of Kenya are maize, wheat, and rice, which require advanced agricultural practices to attain high yields. Phytohormone search for cytokinin and gibberellin queries reflect the role that agricultural production plays in their society and its contribution towards the GDP of Kenya. Ethylene queries, however, were dominated by Asian countries, namely South Korea, followed by Singapore, Malaysia, Qatar, India, and Pakistan. Ethylene is a significant product that is made by chemical industries, and South Korea and Singapore are highly industrial. Hanwa Total, a petrochemical company, increased its ethylene production capacity by 30% in 2017 [57]. Lotte Chemical estimated its ethylene production to be 200,000 tons per annum [58]. These chemical and petrochemical companies in South Korea may represent the major contributing factor for the highest number of ethylene queries globally. South Korea also had the greatest number of jasmonic acid queries globally. Jasmonic acid is associated with pest control, and hence jasmonic acid is used as a seed treatment for pest control in germinated seeds. The application of jasmonic acid induces the production of a protease inhibitor in plants that protects plants from insects [59,60]. The high number of jasmonic acid queries in South Korea may reflect the desire of people, farmers, and researchers in that country to produce pesticide-free crops. Iran and Japan also appear to be following the same principle. India had the greatest number of abscisic acid queries globally. Approximately 18% of India’s GDP is generated by the agriculture sector and provides 50% of the nation’s employment [61]. A major portion of the Indian economy is dependent on agriculture, and the high number of queries on all nine phytohormones may reflect the role agriculture plays in the economy of India. Similar relationships can be stated for Malaysia and Pakistan. These latter countries are developing countries and their economy largely depends on the agricultural sector. The greatest number of abscisic acid search queries was concentrated in the countries of the South Asian sub-continent. Abscisic acid triggers the closure of stomata and is, thus a signal which can minimize transpiration. Farmers often use applications of abscisic acid to reduce the rate of transpiration and avoid drought stress. As India, the Philippines, and Thailand are tropical and sub-tropical climates, as well as Australia, to some extent, the crops in this region exhibit high transpiration rates and, thus abscisic acid use occurs maximally in this region to limit drought stress and maintain crop productivity. China had the greatest number of brassinosteroids queries globally. Notably, brassinosteroids play a prominent role in the physiology of horticultural crops. BRs have the potential to increase the quantity and quality of horticultural crops and also increase stress tolerance [62]. Therefore, it is considered as a plant “strengthening substance”. The application of brassinosteroids may contribute to making China one of the top five global producers of buckwheat, maize, millet, rice, wheat, potato, sugarcane, grapes, apple, sunflower, orange, and banana. Similarly, but lower numbers of brassinosteroids queries were observed for India and the United States. China also had the greatest number of strigolactones queries globally. Strigolactones play a critical role in establishing the symbiotic relationship between arbuscular mycorrhizae and the roots of their plant hosts. The symbiotic relationship provides nutrients, especially phosphate, from the soil to plants. Therefore, arbuscular mycorrhizae are currently used to a large extent in animal manures used to fertilize plants and help enhance crop productivity in sustainable agricultural systems. China is focusing more on the use of organic composts in sustainable agricultural management systems to produce organic green vegetable crops [63,64]. In addition, strigolactones are also used for seed germination [65,66]. Being a larger agriculture producing country, China might be using strigolactones to enhance the seed germination potential of the seeds. The Philippines had the greatest number of salicylic acid queries globally, followed by Trinidad & Tobago, Singapore, Nepal, Canada, the USA, Australia, and others.

## 4. Conclusions

‘Google Trends’ gives us geographic information of different countries who have searched a particular term in Google. Data on the information-seeking activity of people in different countries provides a wealth of useful knowledge on the needs, economy, health, and trending prospects of individual countries. Data obtained on phytohormone queries in the developing country of Zambia is a classic example of how unexpected trends can be observed. The study indicated that African countries have a greater interest in searching for information on phytohormones than American or European countries. This study can be utilized further by the Food and Agriculture Organization (FAO) and other organizations that are involved in the development of agriculture practice in African and Asian subcontinents. Although Google trend search is a valuable search engine, it also suffers with a few limitations. Google trends only compares the keywords rather than providing any objective indication on the absolute number of popularities of a particular term. On Google trend, we cannot find exactly how many people searched for a particular word in a specified time period, thus provides a relative number of searches. Some search terms might not be displayed due to lower search quota, which might not be negligible for absolute number. The statistical robustness of the Google trend data can be increased by increasing the repetition of searches. The spatial and temporal scales also represent the magnitude of the searches, and hence the limitation can be acceptable.

## 5. Materials and Methods

### 5.1. Data Collection

The current study was conducted using time series data on query searches beginning from the start of the time that Google began to collect such data (1 January 2004) to the present time (20 October 2019). It resulted in Google trend results according to different countries over the mentioned time period. We downloaded queries made separately on nine different phytohormones using ‘Google Trends’ and downloaded the data as an csv file, and then merged all of the data into a single Excel file to conduct a comparative analysis. The analysis of nine phytohormone queries included abscisic acid, auxin, brassinosteroids, cytokinin, ethylene, gibberellin, jasmonic acid, salicylic acid, and strigolactones. Each plant hormone was searched individually using the exact text as mentioned, and neither alternative or any synonymous term, was used in this study. We analyzed the data on each of the phytohormones based on country. Each csv file covered 253 countries, but only data from countries that showed a significant number of search queries in the Google search engine were included. In each csv file of the data, 100 was used to represent the highest number of queries, while 0 represented the lowest number of searches in Google. Google trend only provides relative data and does not provide any absolute search data, and hence all the study explained in the manuscript is associated with the relative data.

### 5.2. Data Normalization

Increases in the number of search queries increased its own average over a period of time and hence served as the denominator for later comparisons. This reduced the sensitivity in the detection of changes in search trends based on search volume. ‘Google Trends’ controls the bias created by unrelated common search queries. For example, the search term “disease” may be normalized by dividing its’ number by the search volume of the unrelated search term, “music”. Normalization also compensated for the population size of a region, thus making it possible to rank the countries purely on the basis of search volume. For example, if the proportion of the Indian population using the query term “music” is the same as the proportion of Bangladesh, the effect of the larger population of India was factored out.

To rank the top countries, ‘Google Trends’ takes a sample of all search queries made on the web and determines the area from which most of the queries originated. The internet protocol (IP) address was used to establish the origin of the web queries by analyzing the server logs. An algorithm later calculates the ratio of a variable’s search volume from each country, and the total search volume from each country.

### 5.3. Statistical Analysis

Time-wise and country-wise ‘Google Trends’ data were collected for each of nine individual phytohormones. To analyze the trends for each of the different phytohormone search terms, monthly data for all of the phytohormones were place in an Excel file (MS Excel 2016) and used to generate a line graph. The resulting country-based search data were used for principal component analysis (PCA) using freely-available Past3 software (https://www.techworld.com/download/office-business/past-314-3330821/). Multivariate linear regression analysis was conducted using Past3 software. A variance-covariance model was used to construct the PCA plot with 1000 bootstrap replicates. The same file was later used to construct a cluster graph to illustrate closely related/grouped countries for each search term using the paired group UPGMA method with the Euclidean similarity index, and 1000 bootstrap replicates. A stacked bar plot was also created using the Past3 software. A frequency plot and statistical analysis of all the nine plant hormones were conducted using the Prism software version 6. The graphical map of search queries for each of the nine different phytohormones was downloaded from ‘Google Trends’.

A Venn diagram was constructed in order to illustrate which countries were associated in common with a phytohormone search term. The Venn diagram was constructed using InteractiveVenn (http://www.interactivenn.net/) software [67].

## Figures and Tables

**Figure 1 plants-09-01248-f001:**
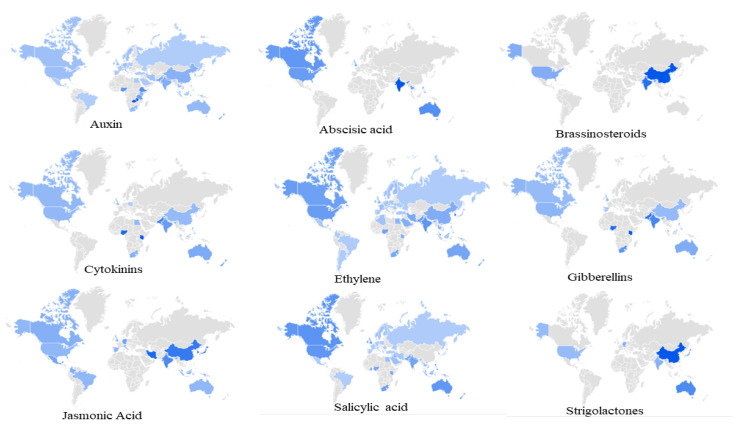
Global map of phytohormone search. The deep blue color indicates a high level of search query for the respective phytohormone, while the light blue color indicates a low search query. The map indicates data from 1 January 2004 to 20 October 2019.

**Figure 2 plants-09-01248-f002:**
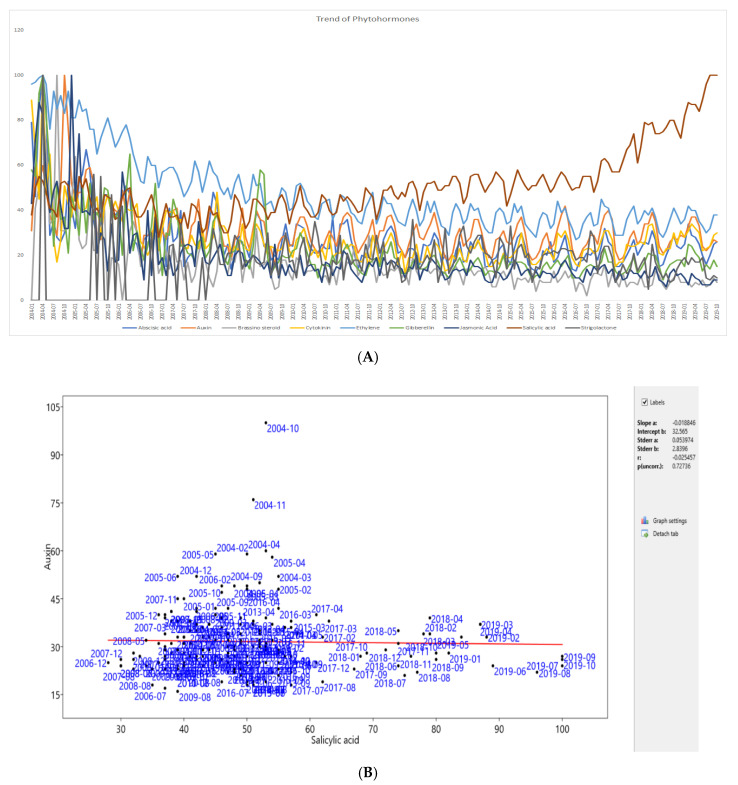
(**A**) Trends of phytohormone search queries from January 2004 to October 2019. The trend shows the search query for salicylic acid is continuously growing compared to other phytohormones. The search query for strigolactones was very low. In total, nine phytohormones, namely abscisic acid, auxin, brassinosteroids, cytokinin, ethylene, gibberellin, jasmonic acid, salicylic acid, and strigolactones were taken into consideration. (**B**) Regression analysis of salicylic acid with auxin showed negative correlation (y = 51.708 − 0.034x, slope *a* = −0.018, *r* = −0.025). The regression analysis of salicylic acid with other phytohormones are provided in Appendix A.

**Figure 3 plants-09-01248-f003:**
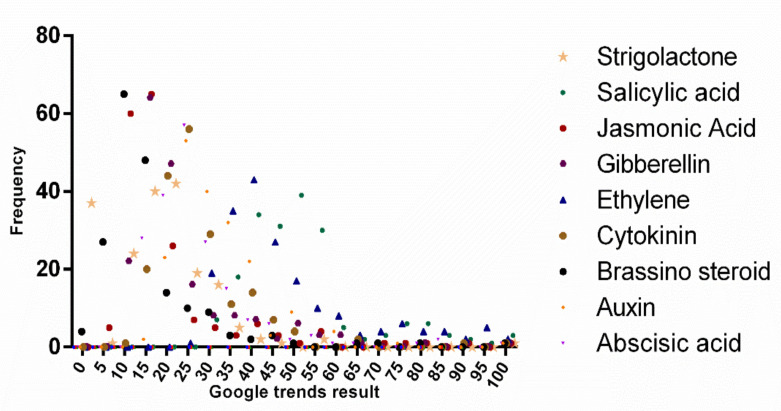
Frequency distribution of different phytohormones. Google trend search data over time was considered to draw the graph. The graph shows various phytohormone search results of Google trends found dynamically. However, a very few numbers of (~<20) Google trend results were found with a score more than 35.

**Figure 4 plants-09-01248-f004:**
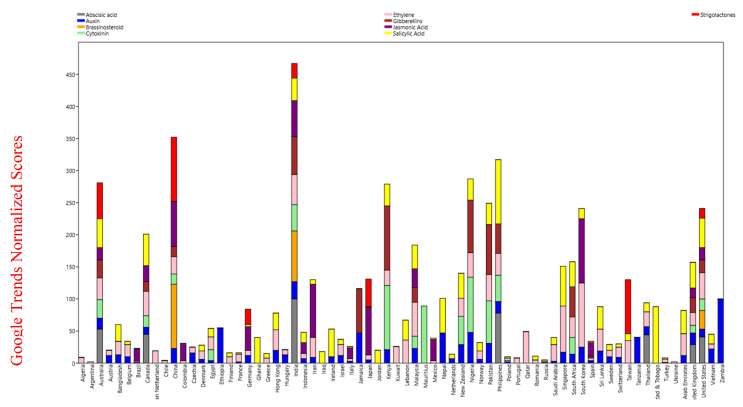
Stacking and area chart of phytohormone search of few countries. The data that resulted from the Google trend search was considered to draw the graph. From the figure, it can be observed that only India and the United States of America show nine stacks describing the search query for all the nine phytohormones. Australia has eight, China and the United Kingdom have seven, and the Philippines has six stacks, showing the respective phytohormone searches in these countries.

**Figure 5 plants-09-01248-f005:**
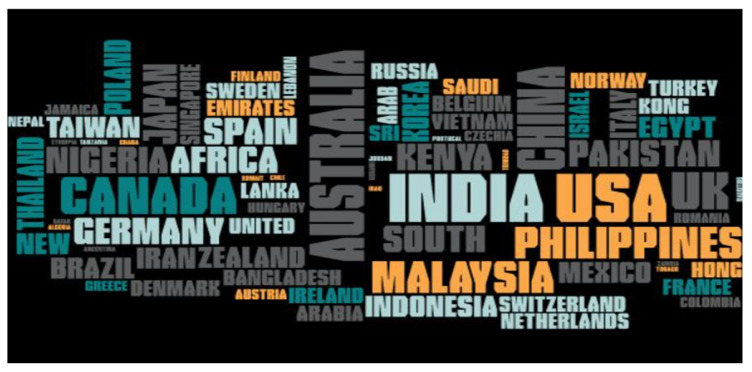
Word cloud of phytohormone search countries in the Google. The word cloud shows the most phytohormone search countries of the world. It includes India, the USA, United Kingdom, the Philippines, Australia, Kenya, Canada, and others. The word cloud was generated using the Google trend search for phytohormones for different countries.

**Figure 6 plants-09-01248-f006:**
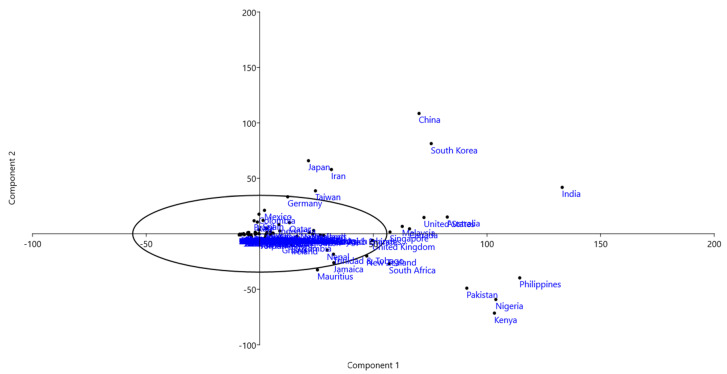
Principal component analysis (PCA) of all the countries associated with phytohormone search queries of all the nine studied phytohormones. PCA show India, United states, Australia, South Korea, China, Malaysia, Japan, etc. fall distantly in one coordinate whereas the Philippines, Nigeria, Kenya, Pakistan, the United Kingdom, South Africa, New Zealand, etc. fall distantly in another coordinate. The countries those fall distantly in the PCA plot show search queries for a greater number of phytohormones while those present in the center has none or zero result. The variance-covariance matrix was used to draw the PCA plot with 1000 bootstrap replicates. The Eigenvalue and variance of the PCA plot can be found in the Appendix A.

**Figure 7 plants-09-01248-f007:**
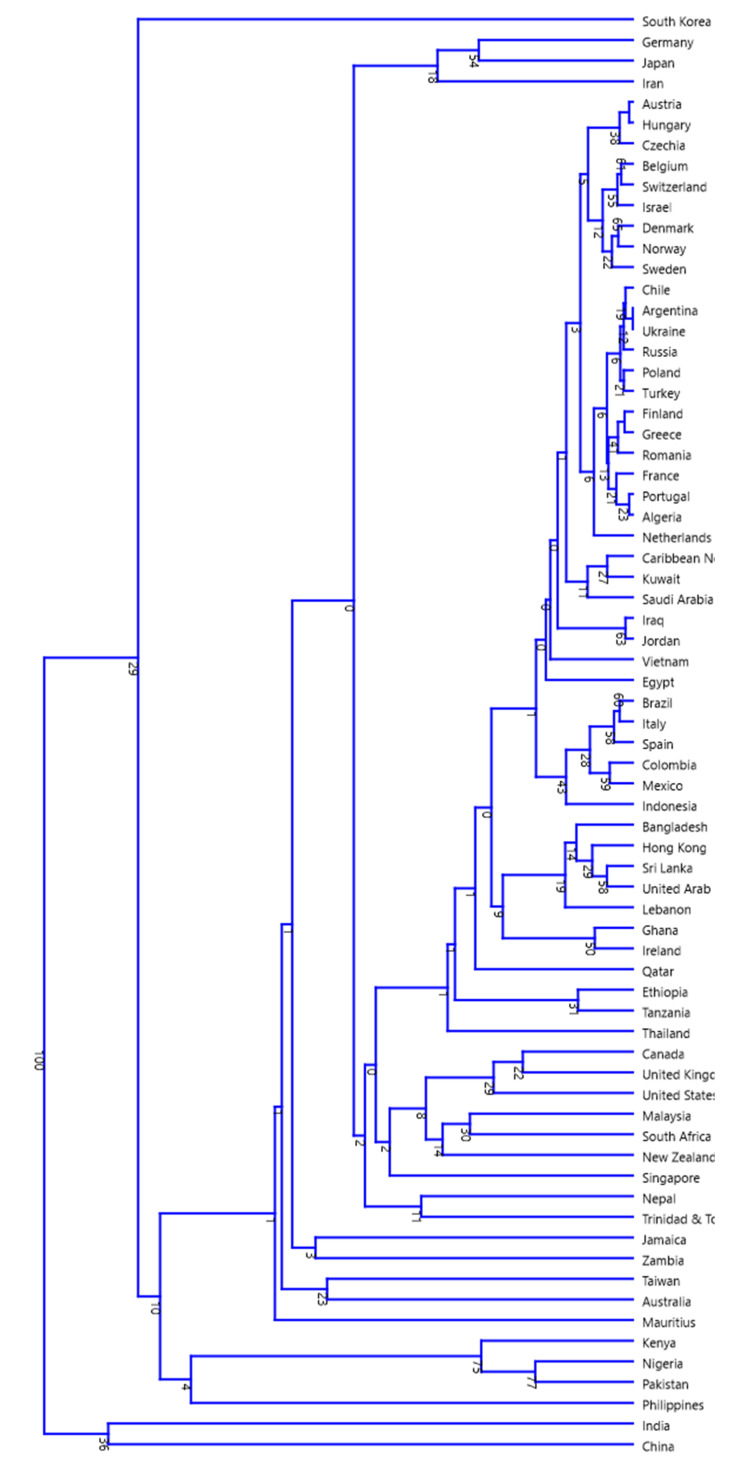
The neighbor- joining tree of the countries associated with phytohormone search. In the tree, India and China grouped together in one cluster, whereas South Korea fall alone in the other cluster. The African country Kenya and Nigeria grouped with Pakistan. Similarly, Germany, Japan, and Iran also fall in one group.

**Table 1 plants-09-01248-t001:** Top five crop-producing countries of the world [41].

Crops	Top 5 High Crop Producing Countries
Apple	China, United States of America, Poland, Turkey, India
Barley	Russia, Germany, France, Ukraine, Australia
Buckwheat	Russia, China, Ukraine, France, Poland
Coffee	Brazil, Vietnam, Colombia, Indonesia, Ethiopia
Grapes	China, Italy, United States of America, France, Spain
Maize	United States of America, China, Brazil, Argentina, Mexico
Millet	India, Niger, China, Mali, Nigeria
Oat	Russia, Canada, Australia, Poland, Finland
Orange	Brazil, China, India, United States of America, Mexico
Potato	China, India, Russia, Ukraine, United States of America
Pulses	India, Poland, Mozambique, United Kingdom, Pakistan
Rice	China, India, Indonesia, Bangladesh, Vietnam
Rye	Germany, Russia, Poland, Belarus, Denmark
Sorghum	United States of America, Nigeria, Sudan, Mexico, Ethiopia
Triticale	Poland, Germany, Belgium, France, Russia
Sugarcane	Brazil, India, Thailand, China, United States of America
Sunflower	Ukraine, Russia, Argentina, China, Romania
Tea	China, India, Kenya, Sri Lanka, Vietnam
Wheat	China, India, Russia, United States of America, Canada

**Table 2 plants-09-01248-t002:** Statistical details of phytohormone search over different time.

Statistical Parameters	Abscisic Acid	Auxin	Brassinosteroids	Cytokinin	Ethylene	Gibberellins	Jasmonic Acid	Salicylic Acid	Strigolactones
Number of values	190	190	190	190	190	190	190	190	190
Minimum	13.00	16.00	0.0	12.00	27.00	9.000	6.000	28.00	0.0
25% percentile	20.00	25.00	8.000	21.00	36.00	15.00	11.00	40.00	9.000
Median	25.00	29.00	12.00	24.00	42.00	18.00	14.00	48.00	17.00
75% percentile	30.25	37.00	17.00	31.00	53.00	25.00	20.00	55.00	22.00
Maximum	100.0	100.0	100.0	100.0	100.0	100.0	100.0	100.0	100.0
Mean	27.56	31.61	15.80	27.53	47.68	23.16	18.96	50.62	16.51
Standard deviation	12.90	10.63	12.87	11.59	17.00	14.47	14.57	14.37	12.52
Standard error of mean	0.9357	0.7715	0.9338	0.8410	1.234	1.050	1.057	1.042	0.9081

**Table 3 plants-09-01248-t003:** Venn diagram result of the common countries of phytohormone searches with different phytohormone combinations. Six phytohormones, auxin, abscisic acid, cytokinin, ethylene, gibberellins, and salicylic acid were taken into consideration during this study. Phytohormone auxin was found in Zambia, Ethiopia, and Tanzania, whereas cytokinin was found only in Mauritius as these countries were not found in the search results for one or other phytohormones (e.g., gibberellins, cytokinin, ethylene, etc.).

Phytohormones	Common Countries
Auxin	Zambia, Ethiopia, Tanzania
Auxin and gibberellins	Jamaica
Cytokinin	Mauritius
Salicylic acid	Trinidad & Tobago, Ghana, Jordan, Iraq
Ethylene	Qatar, Kuwait, Algeria, Portugal, Colombia, Chile, Argentina, Ukraine
Ethylene and salicylic acid	Lebanon, Finland, Greece, Romania
Auxin and salicylic acid	Nepal
Auxin and ethylene	Czechia, Hungary, Austria
Auxin, cytokinin, ethylene, gibberellins	China
Abscisic acid, auxin, ethylene, and salicylic acid	Thailand
Abscisic acid, auxin, cytokinin, ethylene, gibberellins, and salicylic acid	India, Philippines, Australia, Canada, United States, United Kingdom
Auxin, cytokinin, ethylene, gibberellins, and salicylic acid	Nigeria, Pakistan, Malaysia, Kenya, South Africa
Auxin, ethylene, gibberellins, and salicylic acid	Spain
Auxin, cytokinin, ethylene, and salicylic acid	New Zealand, Egypt, Poland
Auxin, ethylene, and salicylic acid	South Korea, Vietnam, Hong Kong, Sri Lanka, Singapore, Taiwan, Bangladesh, Israel, United Arab Emirates, Germany, Ireland, Sweden, Belgium, Iran, Switzerland, Indonesia, Netherlands, Denmark, Norway, Japan, Saudi Arabia, Italy, France, Mexico, Brazil, Turkey, and Russia

## Data Availability

All of the data utilized in this study were obtained from the publicly-available ‘Google Trends’ search engine. Data associated with the manuscript is provided in Appendix A.

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
