# Peer review of "Global Trends in Phytohormone Research: Google Trends Analysis Revealed African Countries Have Higher Demand for Phytohormone Information"

_plants, 2020, doi:10.3390/plants9091248_

Round 1

Reviewer 1 Report

The manuscript “Global Trends in Phytohormone Research: A Google Trends Approach” has a very intriguing title.

Authors have used facilities of “Google Trends” to discover intensity of publishing on such an important subject in plant biology as phytohormones. For the first time it was combined with allocation of these investigations over the globe. The period of the interest was started since January 1, 2004 and ended in October 20, 2019. The study focused on 9 phytohormones has been investigated for decades and according to presented data is still of interest in 253 countries. The overview made an attempt to link the intensity of hormone investigation with a necessity of intensification of agriculture. It is supposed that the knowledge about hormones gives an advantage for countries with high crop productivity.

However, after deep acquaintance with the manuscript, it is very complicated to answer a very simple question: What is a conclusion??? Wide variety of plant hormones investigation over different countries is well known. It is based on the progress of so-called scientific schools. Those scientists, who are working in the field of plant hormones, know modern approaches of these laboratories or groups. This historical background is, probably, the reason that all 9 hormones are now discovered in USA, eight in Australia, 7 in UK. Intensive development of economics and even more intensive collaboration of scientists from these countries and such developing countries as China, India and etc. could be the reason of elevation of plant hormone research therein.

Nevertheless this manuscript looks like newspaper study. The only difference is application of math modeling for big massive of data that was obtained through the Internet. I think that at present state the manuscript could not be published in such scientifically developed journal as “Plants”. The presented material should be improved with better characteristic of modern development of agriculture and modern demands of plants via processes regulated by phytohormones in countries that have the highest interest in hormone research. Thus, at present state the manuscript has to be rejected.

Author Response

Dear Editor,

Please find the response to reviewer comments as an attached file.

Regards

Reviewer 2 Report

General Comments

In the materials and method section, the authors describe the Google Trend data. Importantly, the data are pre-normalized for population. This might be skewing the data due to internet availability. For instance, countries like Zambia may have only concentrated internet availability (53% of the population) in industrial or academic sectors, thereby concentrating search terms by subject. Conversely, countries with wide-spread internet like those in western Europe (~95%+ of the population) and the USA (89% of the population) may have “diluted” the same Google Trends terms used in industrial or academic sectors. I would argue that pre-normalized data would be better to report.

While I love the artistry of Figure 3, it is a bit overwhelming and I am unsure it conveys the information effectively. I am unsure what it shows that is different from Figure 4.

I am confused by Table 1. It would seem like it reports countries that share combinatorial phytohormone queries. If that is so, why is there single phytohormone entries? (e.g. Auxin has Zambia, Ethiopia, and Tanzania). That data was presented in Figure 1. Additionally, there are several countries indicated in Figure 1 that are not reported in Figure 1 under single-hormone entries. I’d recommend the use of lines or gray shading to help delineate the rows in this table.

Line-by-Line

Line 21: Application of phytohormone research? Or do the authors mean the actual application of phytohormones on plants?

Lines 36-37: I’d recommend getting rid of one of the “plants” used in this sentence.

Line 49: “inhibit” should be “inhibits”.

Line 56: It looks like there is an extraneous period-mark here.

Line 56-61: In the context of agriculture and land management, I think the authors should consider the use of synthetic auxins (specifically, 2,4-D) as herbicides.

Line 86, 94, and 103: Here, the authors start each paragraph about three additional phytohormones with their commonly used abbreviations. Other paragraphs start with the fully written form. I’d like to see these three paragraphs start out with the full-form of the hormone name.

Line 92093: This statement needs a citation.

Line 98: The statement about plant-plant communication needs a citation.

Line 99: The statement about JA’s role in wounding needs a citation.

Line 102: The line about the JA/Ethylene balance in apical hook formation requires a citation.

Line 110-111: This statement needs a citation.

Line 113: The genus and species names should be italicized.

Line 117: The statement about phosphate sensing needs a citation.

Line 124: The statement about agricultural uses of phytohormones should have a citation.

Line 126: Citation needed

Line 128: Citation needed

Line 130: Consider citation, though not as necessary as the others I have mentioned

Lines 137-176: Why is this text bolded?

Lines 144-157: This text is a bit daunting to the reader. Is there a better way to summarize this? I understand that the data is already published and so the authors cannot readily summarize the data in a table, but perhaps it would be beneficial to leave citation 132 as the end of the following sentence?

Line 166: While I don’t doubt it, this statement should be backed up by a reference. Alternatively, the authors can state that “one of the means by which people all over the world search the internet for agricultural information is Google.”

Line 179: The word “examined” is vague here. Perhaps “queried” is better?

Line 180: It would be beneficial if the authors informed the reader exactly what search term was queried. As their introduction alludes to, there are many synonyms for phytohormones (e.g. “auxin”, “IAA”, “indole-3-acetic acid” and “indole-acetic acid”)

Line 184: What is the temporal aspect to this figure?

Line 191: While I recognize that the authors explain what the blue coloration means, there should be a color-scale indicating what the coloration means. What does “high level of search query” mean, exactly?

Line 239: The y-axis on this plot has no label or units. This concern connects back to my general comment made above.

Line 279: The y-axis on this plot has no label or units.

Like 378: This statement should have a citation.

Line 382: The authors state that the increasing number of SA queries “reflects the extensive use of SA derivatives in the cosmetic and health care sector”. But the presented data does not show that. Could the authors find search terms that are affiliated with SA? Otherwise, they should tone down their language about their conclusion.

Author Response

(The authors gave the same response as above.)

Reviewer 3 Report

GENERAL COMMENTS:

I am not sure what the study is trying to do or show. Google Trends is a very interesting analysis tool and I can see so many interesting ways of using it. Unfortunately, this study while aiming to “assess ‘Google Trends’ using the phytohormone search terms”, has no significance at all. It seems like this study has never been properly developed and needs complete redoing to have an interesting angle.

Major general concerns include:

The rational is totally unclear: The authors introduce the extensive biology for phytohormones, which is totally irrelevant for this study. Authors introduce which country is the biggest crop producer and then not use it at all for the analysis. Somehow this then means need to look at trends for phytohormone searches. I cannot follow this at all. 

I cannot see any relevant results. The data is poorly analyzed and described. The statistical methodology is misinterpreted and/or incomplete at many spots.  

The methodology is very unspecific, underdeveloped to the point where none of this work is reproducible. The methodology used is often not appropriate. There are many other text-based strategies that would suit much better.

There is no discussion since there is not results. The discussion reads like an introduction. In fact, it would serve as a better introduction to this subject.

The conclusions reads as the first line of the results section displaying the only shallow result the authors have provided.

This study needs to be completely redesigned. Some specific comments to the authors below:

SPECIFIC COMMENTS:

Introduction

Line 47 - 118: The description of the hormones can be shortened to one paragraph or deleted altogether if this is not looked at on a biological basis. It might fit in the discussion if there is a trend paired with an agricultural application of the hormone. Unless the biological function is of relevance to the paper please leave this out as it  distracts.

Why is it important to inspect the trend?

Line 137: fix the format (unbold)

Line 143 – 157: These facts would be better off in a table. It is very distracting as a text chunk

Line 157 – 176: The authors, I assume, are producing their rationale here why it is important to look at trends. I cannot see any relevance of the google trend in respect to agricultural differences in countries around the world. The authors assessing the trend per country later, but fail to give this relevance towards agricultural practice etc.

The introduction is missing a description of the capability, purpose and limitations of the search engines/tools used as well as what makes it necessary to use Google Trends versus other search engines.

Results

Fig2 is very hard to interprete and could be displayed better. Is there a trend? Can the authors use shifting averages to smoothen the curves? Most of them look like they are going down overall! There is plenty of strategies to analyze this better. Please use proper statistics and a better visualization. I do not understand what the actual result is here that the authors would like to display.

It looks like there is a seasonal trend as well? Have the authors taken that into account?

How do the actual numbers of the searches compare between phytohormones? Which one is most popular?  Can the authors compare them to each other?

Figure 3. Not sure what the relevance is of this? Why is it important? Unless the authors correlate it to some climate, (agri) cultural parameters that countries share or not share. I would leave this out and go straight to PCA? It also doubles up with data in table 1.

Figure 4. Since the numbers are relative, I am not sure if this graph tells the reader anything. It seems misleading towards proportions of searches because as far as I can tell the difference search queries are not normalized to each other? The legend is not sufficient in telling the reader what this data means.

Line 292: Please specify the type of clustering analysis used for the PCA results or specify that this was not done here but in a hierarchical cluster. What was the method and parameters. What was the aim of this analysis? And please display this in the graph. There seems to be clusters but what is the confidence of these?

Fig 6: The cluster of “India and China” is less the non-convincing in the PCA! In figure 7 those two countries may fall beside each other on paper but the distance between them (branch length) is rather extensive compared to other clusters in the tree. The authors are plain wrong here. Please improve the tree visualization and use this method appropriately.

Do clusters correlate to crop types grown in these countries?

Apart from the crude data output, there is no interesting results given by the authors.

Discussion

Line 312 – 380: (The whole discussion) This is what needs to be the introduction. This sets up the rational for why you are looking at using Google Trends. It needs improvement and clarity if the authors wish to use it as an introduction.

Line 380-384: Is what you would expect in the results to a much greater extend (not just one sentence).

There is no discussion and the conclusions have nothing to do with the aim of the study, the results shown nor anything discussed.

Material and methods

Line 400:”The analysis of nine phytohormone queries included abscisic acid, auxins, brassinosteroids, cytokinin, ethylene, gibberellins, jasmonic acid, salicylic acid, and strigolactones.” Please specify or clarify the exact search terms in the methods or supplementary. This way results can be reproduced.

Line 403: “only data from countries that showed a significant number of search queries in the Google search engine were included” Please define “significant” in this case. Please specify the Google engine (Google Trends? Or Google)

Line 404 onwards: The search data is relative. Is there a way of absolute measures to get an idea how many searches there were over the years? (20 searches versus 20,000?) from the described normalization of Google Trends? This would make a difference in sense of significance of these hormones!

Statistical analysis: Countries where clustered towards their similarity of searches across the 9 phytohormones? Please specify sufficient methodology and parameters used for the analyses.

Author Response

Dear Editor,

Please find the responses to reviewer comment as an attached file.

Regards

Round 2

Reviewer 1 Report

I have to admit that authors made an attempt to improve the manuscript “Global Trends in Phytohormone Research: A Google Trends Approach”. They answered previously raised questions, rearranged manuscript data and changed the conclusion.

Nevertheless I still have several comments and the most important is following:

Google Trends, a comparatively newly developed Google Inc. portal, which freely accessible for absolutely diverse searchers and which accumulates a wide spectra of information of various kinds. It permits to generate data on geographical and temporal patterns according to specified keywords. In several publications that were mentioned by authors, there are examples of useful applications of Google Trends in health care, especially in the field of epidemiology. But even there it was estimated that Google Trends has modest reliability in case of minor media coverage and that Google Trend data may contain inaccuracies.

As I understand the aim of presented in the manuscript investigation was to prove facilities of this tool to analyze data on plant biology, especially in the field of the public interest in plant hormones. Authors provided a wide analysis of interest in 9 phytohormones in 253 countries for 15 years. Finally it was concluded that countries are quite different in the interest to various plant growth regulators. Surprisingly there were estimated countries with mono-interest, for example, like Zambia, as well as countries with interest in all 9 tested phytohormones. Hence it is very complicated to understand the reason of such attention. Authors speculate about it but the explanation couldn’t be taken without a doubt. For example, the interest in strigolactones in China was suggested to relate to the interest in plant-microorganism symbiosis. But agricultural application of fertilizers based on mycorrhiza nowadays is widely spread all over the world. So, why it is China? Also it is very complicated to conclude about the origin of these queries. Do it directed from researchers, farmers, students or just reflects interest rose from mass media??? Probably it might be useful to get some data from already well known open access data-base as PubMed or Agricola and compare them to each over. In this case it would be of importance to choose one phytohormone and to check number of publications on this subject. Otherwise this investigation looks closer to sociological test and do not reveal the reliability of Google Trends application in plant biology.

Taken together with quite a number of notes concerning text and tables design I have to conclude that the manuscript still could not be published at present state.

Author Response

Dear Reviewer

Thanks

Reviewer 3 Report

Please be advised that "general comments" are a general view of the study and specific examples of these critics are given in detail in the "specific comment" section. I also would like to point out that I don’t have to give specific methodology references or solutions to all my critic when there is so much not developed yet. This would be equivalent to designing the authors study from the point of a conceptual idea, which is not a purpose of a revision. That said I hope this extensive, more than usual effort, will help the authors to complete this study. Please spend sufficient time (much more that 1 week) to address scientific issues and pay attention to all details. Maybe this study can be presented at the local institution for feedback prior to resubmission to scientists not in the field as well as in the field. 

GENERAL COMMENTS

The study aims to investigate the importance of phytohormones in different countries by researching “global trends associated with “phytohormone” queries” . The rational according to the authors states that it is important to boost agricultural productivity, which varies a lot between countries.

While I understand the overall aim better now with the improved manuscript, I think the study and the manuscript requires substantial work before being sound overall. The manuscript is extremely hard to follow and the data still not visualised to the point that the reader can extract from the graph what is said in the results let alone being independently understood. There is not logic flow to this manuscript and I am left with little new insights and too many questions.

The results are very confusing and it took a long time to assemble the pieces to assess the discussion properly. There are 3 sections. (1) an insight into which country searches what hormone, (2) yearly trends of each hormone globally, (3) similarity in search patterns between countries (what time frame). The authors also need to revise the vocabulary used. It is confusing what the word trend is used for. Are the authors talking about trend over time or differences between countries? These are two different aspects of this study and they get interchanged. This needs to be clear as it was impossible to follow.

I actually would start the results with section 2 using all countries combined over time-trend. This is a better introduction to the subject where the global trend is assessed over the time frame. This was interesting even though visualisations could be simplified a lot. Unfortunately, this trend was not discussed or explored why that is. If I missed it then it was not prominent/clear enough and it will not be clear to the majority of the readers. This trend e.g. could be discussed in the first paragraph of the discussion specifically.

Results after the annual trend can then follow for assessing which country contributes to the global trend (are all countries increasing for SA?). Then this follows into how countries differ. At the moment it switches between countries and years and back to countries.

Since there is so much data to draw together (climate, economy, land use etc) the authors could work a lot more with associations and tables giving overview of these things but without duplication of other results. This is currently a big problem in the manuscript.

The results are not convincing and a lot of the methodology is not described (in the methods) enough and not introduced in the results enough to be convinced. The feeling here is that the authors do not understand these methods used. Whether that is true or not is beside the point as other readers will also think this way (maybe not all but at least half). Some of the methodology was justified in the response to the reviewers but I haven’t found these important details in the methods themselves. Whatever is being responded to in the reviewer comments needs to be also worked into the manuscript otherwise it is not improved. Please go through the previous comments and make sure they are all addressed “in” the manuscript as well.

The discussion is a lot better. I am missing some punchy conclusions and some bits missing as detailed below.

SPECIFIC COMMENTS:

Title: Can the title be more enticing? Maybe add the main conclusion into the title. Two main things were the increasing trend of SA and why that is, or that African countries have a higher demand for information?

Abstract:

Still missing the main conclusion of the study here: What does it mean that only India and the US searched all queries? Is there any speculation at least? In the reviewer’s responses the authors mentioned a number of interesting conclusions about how agriculture in countries could have influenced the trends. I am missing this in the actual manuscript/ abstract.

Introduction:

Still a little bit repetitive in the first two paragraphs (line 37-67). Could be shortened to what is relevant to the Google trend research questions directly.

Table 1 got a bit jumbled (format issue while uploading I guess)

Line 102 – 116: What are “traditional data surveys”? I probably would rather mention other means of searches. So how much does Google capture? What are the alternatives? Bing? Going to a library? Is there an estimation how representative this is of all people in the world searching? This does not need to be in the introduction in particular but would help understand the results later.

“In addition, the use of  ‘Google Trends’ enabled us to survey weekly, monthly, and yearly use of phytohormone search queries without any cost” I would reword this as it does cost you your time (= money). Do you mean it is a very cost-effective way of studying trends of phytohormone enquiries?

“in different countries and globally” is the same. Just use “globally”

109: “This analysis will enable us to better understand which countries are querying information on phytohormones and benefitting from their search queries regarding phytohormone research and implantation in agriculture production and which countries are lagging behind and need additional support on specific topics and applications.” This is a long sentence. Please make 2 or 3 out of this one. Hard to follow.

  1. The study aims to understand which countries are querying information on phytohormones? - that is done in the study yes
  2. The study aims to understand which countries are benefitting from their search and implementation in agriculture? - did you address this in the study? How?
  3. The study aims to understand which countries are lagging behind and need additional support on specific topics and applications? – this is the opposite of 2 did this come out of the study. It wasn’t clear enough? If so, you can discuss measures of improving the situation for these countries.

Make sure all these questions above are addressed and discussed in the manuscript.

Results:

Section 1-country-wise differences

Line 120: “naturally-occurring phytohormones” Is there a significant use of non-natural ones? Would this be a relevant counter-trend to assess? This could be in the discussion? E.g. if countries are more open to synthetic products there may be more searches on that?

Line 125: “The countries with the greatest number of queries”. Can the authors add a sentence how this reflects or not reflects the population behind it? Briefly explain what these numbers mean since they are normalised (e.g. compared to unrelated searches as described in the methods).

The suppl. Data should be combined into 1 table.

Fig.1 is pretty but not easy to oversee. Some sort of extra bar graph may be more useful to quickly assess the differences. Look for some visualisation aides.

Section 2: Yearly trend

Line 169: “to analyze yearly trends in phytohormone search queries” refers to fig.2 but fig.2 does not show yearly trend but an every three month trend cycle. A yearly trend would be much better. Can the authors use an average measure per year and compress the figure to one data point per year (try a plain average for each year and have a look). There is too much in there and the legends are too small. All I see is that all searches are going down and are low except for SA. The rest seems speculative.

On that note, why do the authors think the trend is going down for all but SA? Regression stats would be good here!

Line 171: This needs to be explained better. What does this mean in a practical sense? The frequency distribution of what exactly? Please clarify in the results.

Fig.3 legend is insufficient to understand the data. The graph overall is hard to read.

Table 2: What am I looking at here? What does this table tell me? If this is the data summary of the years (all countries?) the authors should rather try a regression.

Results section 3 – commonality of searches amongst countries:

Where is supplementary figure 11?

Line 223: Where is the word cloud?

Fig.5 labels are is too small, consider swapping x/y to make the graph longer and the writing larger. In the legend “From the figure it can be observed that only India and the United states of America shows nine stacks describing the search query of all the nine phytohormones.” I disagree (not with the data per se). I cannot easily see it. I would have to count or compare the column segments. It is extremely slow and hard to see what the authors are claiming. This is not a suitable visualisation for the authors expression. Maybe the authors can use visualisation aides or a different approach to make it easier.

Table 3 and Figs 4-6 show the same kind of result: “which countries share a search pattern” and “query the most hormones”.  I don’t think we need all of these figures, just one good one showing the above 2 statements clearly and the rest can go into supplementary.

Fig.6 is too hard to read. Use numbers and a legend to explain which countries are represented by which numbers. I am missing the cluster statistics and borders of confidence in this graph for the claimed clusters (mentioned in the response to reviewer).Please use visualisation aides. How much variance does component 1 and 2 capture? Please add this to the supplementary or main text. India and China are not a cluster in Fig 6. China and South Korea might be one but we are talking about two data points. Maybe the wording here will concentrate on “they are closer”? For a proper cluster there needs to be more data points to determine a confidence.  

Overall when standing back it is more of a continuous spray towards the component 1 axis. Have the authors analysed which variables contribute to component 1 vs 2 and is there much left in component 3?

Line 260 – 271: Hard to be confirmed when the labels all overlap in fig 6. What do the authors mean by coordinate 1? Component 1 or quadrant 1? Sounds a bit like the latter, but impossible to follow. If quadrant that would make more sense but it does not really tell anything since we are looking at a relative distance between points across dimensions in PCA, not exactly where. The authors should describe the positions as distances along component 1 and 2.

Fig. 7 – labels are cut off and the graphs could be improved using visualisation aids (bars emphasising clusters). South Korea and China are close in the PCA but distant in the cluster analysis? Kenya, Nigeria and Pakistan are a cluster in both. Is there thoughts why? Climate, people, economy, crops in common? A lot more can be explored from this figure but needs research into whether the clustering is reasonable in the real world and not just in numbers.

Discussion:

Line 291: “Therefore, for the first time, we aimed to understand the global trends associated with “phytohormone” queries.” That is a good goal. What have we learned from the trends observed? The study shows that there are trends but no work has been done to understand the increasing and decreasing trends over the years. If this is not discussed, then it should not be in the study at all. However, this is actually an interesting result. Has SA become more important over the last 10 years or more publicised? Is there commercial push that influences the farmers etc? What are the big companies and how badly do they want to sell one over the other product?

Also, why is SA the highest searched? This has not been discussed.

Line 324: “reflects the role” change to “may reflect the role” as this study cannot proof it but only propose the hypothesis.

Line 330 – 332: Tropical climates are rather high humidity and do not have drought issues as much as subtropical. India and Australia are huge and have tropical as well as subtropical. Please revise the climate statements for all countries.

Line 333- 339: What kind of stress? Any? Australia and India would be countries of extreme heat, drought, salt and flooding stress….paired with nutrient sufficiency. Australia in particular does not search for BRs as mentioned in the manuscript. This may not be generalisable.

“China is focusing more on the use of organic composts in sustainable agricultural management systems to produce organic green vegetable crops” Seems irrelevant?

Conclusion:

Conclusions should not introduce new results but rather summarize the discussion and propose an (1) insight, (2) new hypothesis or a (3) changes to current standard procedures.

Delete line 349 – 354 and move the interesting bits (like below) into results or discussion.

 “Data obtained on phytohormone queries in the developing country of Zambia is an example of how unexpected trends can be observed.” Was it? What trend was that and what did it mean? Do the authors refer to SA? This was not discussed.

“data on the Philippines provided information on their potential greater interest in beauty care products.” Move to discussion or results.

“The study indicated that African countries have a greater interest in searching for information on phytohormones than American or European countries” Is a major result not a conclusion. What is the insight from this result? What would be good to happen next? Should be supported? Should research be directed to this geographical area? What benefits would this have? Etc etc etc Some of these questions should be explored or the authors may have their own.

Author Response

Dear Reviewer,

Thanks

Round 3

Reviewer 1 Report

I have to admit that new version of manuscript “Global Trends in Phytohormone Research: Google  Trends Analysis Revealed African Countries Have Higher Demand For Information” was improved. Authors put several accents starting with changes in title. Authors presented a big massive of data and applied different types of statistics and infographics. The most important for me was to convince authors to emphasize that this investigation reflects public interests and not really deals with scientific research. My intention was not to doubt the interest in phytohormones in such countries like India, Kenya, Zambia, etc. But I am little hesitating that Google trends would give a correct information about investigations on phytohormones action.

At present form I can recommend the manuscript for publication.

Author Response

Dear Reviewer,

Thanks

Reviewer 3 Report

GENERAL

The writing has improved and the headings make it easier to read.

The result text reads a better too. The text more often does not match the data and interpretation of the figures.

There is too many figures and too little clarity in the legends.

The study data is poorly present and at that not convincing as it is.

It cannot be assessed whether it supports the conclusions. One of the main results (increased trend of SA) is not discussed. 

Some of the address to the comments were not included into the manuscript and are therefore not available to the future reader. 

SPECIFIC

Salicylic Acid was shortened to SA in the beginning of the manuscript so use SA for the rest of the text. Please check other hormones as well (either abbrev or not throughout).

Introduction

Line 22: Can be shortened to “Phytohormones have been used in agriculture world-wide for many years to improve crop production and continues to be an active area of research for application in plants”

Line 45: change to “Phytohormones are a group”

Line 112: “The “Google Trends” search engine is an excellent tool for assessing country by country trends in the use of search terms searched using the Google browser” I would tone it down a bit since there is a huge issue in Africa in particular with access to the internet due to availability of technology, internet and electricity to the general public. This makes it an uneven measure in comparison to first world countries.

Line 115: “‘Google Trends’ will provide detailed information about the importance of phytohormone research globally.”  I suggest to take out “detailed” since throughout this review process the authors increasingly stated there is some considerable limits to this data in terms of normalisation and resolution.

Results

Please check the format in Figure 1. The colour legend overlaps the writing.

From figure S1 most of the hormone searches are going down since 2004. Is there a reason for this?

The figure 2A and 2B are still very hard to read and interpret.  In Figure 2B, the regression between the phytohormones is confusing. Why didn’t the authors do a regression of SA over the time to prove that the trend of increased searches is significant? The regression analysis shows no slope (slope = 0) so there is no change throughout the time of queries. This contradict the authors text.

Figure S2 is missing or wasn't labelled correctly.

Figure 3 states trends over time but no time line was given in the graph. The figure/data of the figure is not described well in the text or legend. The meaning is not clear.

Table 2 and its data is not well described in the text. It remains unclear what this data contributes to the study.

Suppl. File 11 is missing.

Figure 4 is unnecessary as it doubles up with figure 5. Figure 5 is easier to interprete. Move Figure 4 to supplementary or delete.

Figure 6 looks pretty. The legend is insufficient. What does the size of words stand for? What time frame, what hormones? It cannot stand alone.

Figure 7: is unchanged as well so it is still not showing the clusters the authors talk about in the text. Hard to be convinced without going into the number analysis in the supplement. Even then it is unclear what the cluster numbers refer to in the plot.

Figure 7 contradicts Figure 8.

Figure 8 is unchanged.

Discussion:

Hard to just due to flaws in the result presentation.

Is there a speculation as to why SA is increasing? After all this was one of the main results? It is not mentioned in the discussion.

Author Response

Dear Reviewer,

Thanks
